

# Residential heating emissions for the Western Balkans

Christian Asker[1], Eef van Dongen[1], and Olivier Tasse[1]

[1]Swedish Meteorological and Hydrological Institute, SMHI FoUh, 601 76 Norrköping, Sweden

**Correspondence:** Christian Asker (christian.asker@smhi.se)

**Abstract.** Air pollution adversely effects health, ecosystems and infrastructure. In the *Western Balkans* (Albania, Bosnia and Herzegovina, Kosovo[1], Montenegro, Republic of North Macedonia and Serbia), the air pollution situation is more adverse than in the European Union in general. Understanding the air quality situation requires high quality emission data, with high resolution spatial distribution, especially for enabling remediation efforts.

In this work we have calculated air pollution emissions from heating of individual housing units in the Western Balkan region. The basis for the dataset is a geographical dataset of buildings detected from satellite imagery by artifical intelligence (AI) methods. The building data has been combined with geospatial landuse datasets as well as statistical data for heating needs for residential buildings in the countries included, and finally with emission factors to calculate the heating emissions.

    The resulting datasets provides high-resolution heating emission data for common pollutants and are published as open data

(Asker, 2024) . When comparing national totals for emissions, the datasets in this work are comparable to other, spatially coarser datasets, though the agreement strongly depends on the fuel usage data for each country/region.

## 1 Introduction

The Western Balkans (in this work denoted *WB*) region has longstanding issues with poor air quality (UNEP), especially during the winter season. The region has a population of 17 million (World Bank). While traffic, industries, power production and

residential heating are all considered important sources of air pollution in the region, there is a lack of detailed emission data for all these sectors. Detailed emission data for the most important sectors is vital in order to understand the air pollution situation and in order to be able to improve the situation. High spatial resolution provides not only a more detailed picture of the sources of air pollution, but may improve the quality of atmospheric disperson modeling and help find emission hotspots, which in turn can facilitate effective remediation efforts.

Emission data from district heating facilities is available for at least some of the cities in the WB. Data for heating emissions from individual housing units (1-3 family houses) is very limited however, even though they are excpected to have an important contribution to the total emissions.

    The emissions from individual housing units are typically released at a fairly low height (around roof height) and are therefore not dispersed effectively, especially during cold weather and temperature inversions, leading to a high contribution to

the concentration of pollutants locally. This makes an detailed description of these emissions even more important.

---

[1]All references to Kosovo in this document shall be understood to be in the context of the United Nations Security Council resolution 1244 (1999)



In this work we present a methodology and resulting dataset for emissions from heating of individual housing units in the WB. The emission dataset has high spatial resolution and covers both urban and rural areas. The dataset is compared with datasets for regional (European) scale and suggestions for improving the methodology are discussed.

## 2 Input data

### 2.1 Buildings

The starting point is a building dataset for the WB, from the Global ML Building Footprints project (Microsoft), denoted *ML-Buildings* onwards in this report. For the WB region, the dataset contains about 30 million building polygons. For parts of the Republic of North Macedonia and south-eastern Serbia, there are gaps in the MLBuildings dataset. The gaps were replaced by building polygons from the Openstreetmap project (osm). It is worth nothing that for those areas, the Openstreetmap has fairly low level of coverage of buildings and therefore do not compensate fully for the lack of data in the MLBuildings dataset. An example of building polygons for Novi Travnik (Bosnia and Herzegovina) can be seen in Figs. 1 - 2.

### 2.2 Landuse data

Corine land cover (CLC) (European Environment Agency, 2020a) is a geographic landuse dataset covering Europe, in vector format, having a *minimal mapping unit* (MMU) of 25 hectares. Urban Atlas (UA) (European Environment Agency, 2020b) is a detailed landuse dataset covering european municipalities having at least 100000 residents, with an MMU of 0.25 hectares. These two datasets were merged into a single dataset covering the whole geographical region.

### 2.3 Energy data

The following fuels have been included:

- wood
- wood residue
- pellets
- coal
- lpg (liquefied petroleum gas)
- natural gas
- oil

For some regions only a subset of these fuels have been used since the energy usage data available do not contain all fuels for all regions, nor are all fuels used in all regions.

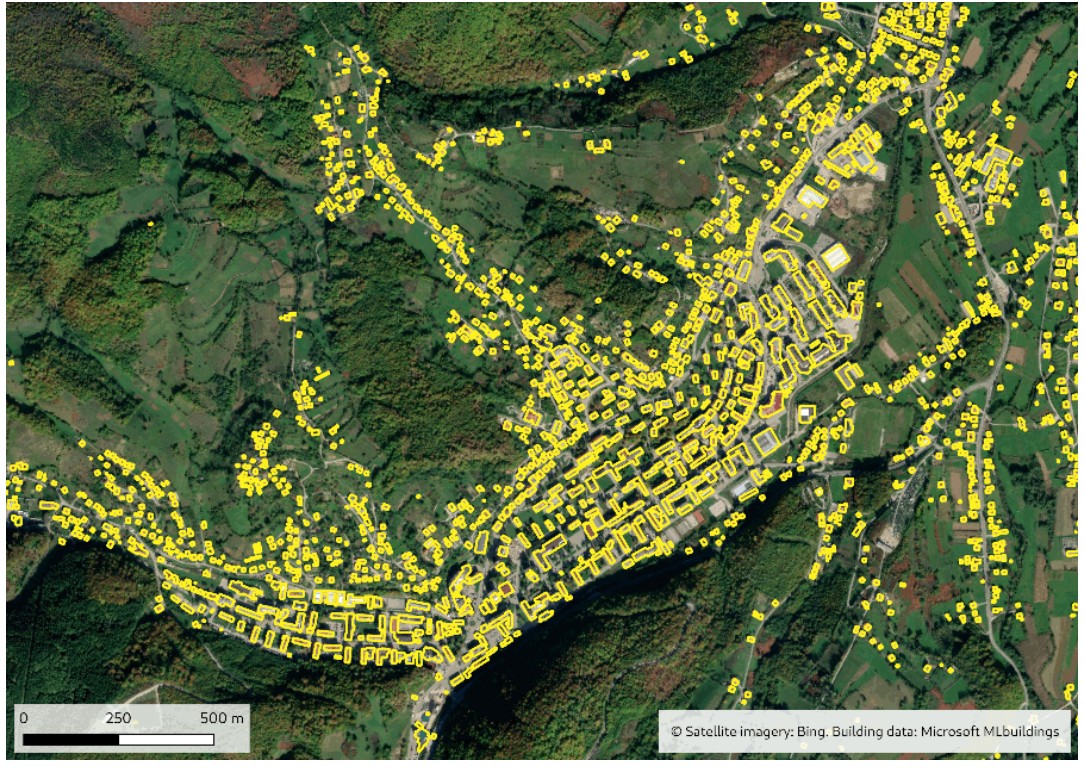

**Figure 1.** Example of the MLBuildings dataset for Novi Travnik (Bosnia and Herzegovina).

## 3 Methodology

The gis operations described below used the following softwares: QGIS (QGIS Development Team, 2021), Python + GeoPandas (Jordahl et al., 2020) and GDAL (GDAL/OGR contributors, 2024).

### 3.1 Filtering buildings

Starting from the MLBuildings + Openstreetmap dataset, the aim was to remove all buildings not 1-3 family housing units.

The buildings are all polygons, allowing for calculation of the building footprint area. Incorrect and duplicate geometries where detected and corrected. Then all buildings having a footprint area less than 50 $m^2$ or more than 200 $m^2$ where removed.

The next step was to merge the landuse data from CLC and UA datasets into a single dataset covering the whole region. The CLC dataset was cutout for the WB region, and the attributes for landuse-classes adjusted to match that of the UA dataset. Further, cutouts were made in the CLC datasets for all cities where UA data was available. Finally the two datasets where

merged into a single one.



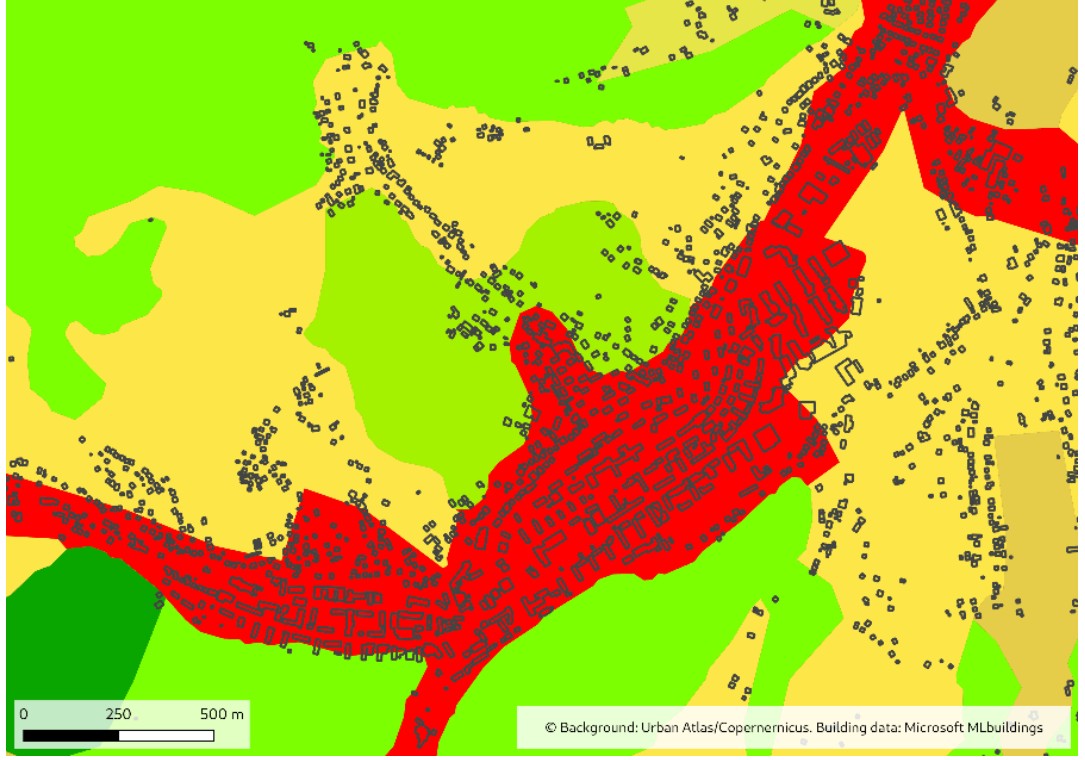

**Figure 2.** Example of the MLBuildings dataset for Novi Travnik (Bosnia and Herzegovina). The map shows the same area as in Fig. 1, but with UrbanAtlas and Corine land cover merged as background.

The merged landuse dataset was then joined onto the buildings dataset, so that each building polygon also had attributes for landuse. The following landuse-classes[2] were assumed to possibly contain som degree of 1-3 family housing units:

- 11100 Continous urban fabcric

- 11200 Discontinous urban fabric

- 11210 Discontinuous Dense Urban Fabric (S.L.: 50% - 80%)

- 11220 Discontinuous Medium Density Urban Fabric (S.L.: 30% - 50%)

- 11230 Discontinuous Low Density Urban Fabric (S.L.: 10% - 30%)

- 11240 Discontinuous very low density urban fabric (S.L. < 10%)

- 11300 Isolated structures

- 21000 Arable land (annual crops)

- 21100 Non-irrigated arable land

- 21200 Permanently irrigated land

---

[2]The classes listed here are a combination of the UA and CLC datasets.





- 21300 Rice fields

- 22000 Permanent crops

- 22100 Vineyards

- 22200 Fruit trees and berry plantations

- 22300 Olive groves

- 23000 Pastures

- 23100 Pastures

- 24000 Complex and mixed cultivation patterns

- 24100 Annual crops associated with permanent crops

- 24200 Complex cultivation patterns

- 24300 Land principally occupied by agriculture with significant areas of natural vegetation

All buildings belonging to landuse-classes other than the above were removed from the dataset. While some of the above landuse classess are mainly agricultural lands, from random sampling of the dataset it can be seen that those landuse classes may contain housing, in particular for areas close to smaller settlements, though the extent of this is different in different regions of the WB region. An example image of the buildings dataset after filtering is shown in figure 3.

Comparing the resulting dataset with statistics in the *Typology of Residential Buildings in Bosnia and Herzegovina* (Arnautović-Aksić et al., 2016) and *National Typology of Residential Buildings in Serbia* (Jovanović Popović and Ignjatović, 2013), showed that the number of buildings in the dataset were still quite large. Especially for landuse categories where there is a considerable mix of housing and smaller farm buildings this is likely to be a problem, leading to a possible overestimation of energy usage for such areas.

      For such areas in Bosnia-Herzegovina and Serbia, by using neareast-neighbour analysis, all buildings that were within 20 meters from an-
other building were selected, for the landuse categories 23100, 24200 and 24300. Of these selected buildings, half were randomly removed. Though these operations reduced the number of buildings for Bosnia-Herzegovina and Serbia, the dataset still contained more buildings than the Typology of Residential Buildings in Bosnia and Herzegovina, which may lead to an overestimation of energy usage and therefore emissions there. This is further discussed in the results section.

For the region surrounding Strumica in North Macedonia, an issue was discovered during the building analysis. In this area, the MLBuildings AI software had detected a large number of large buildings outside urban areas. Most of these were in fact greenhouses and agricultural lands that had been covered with semi-transparent cloth, leading to the errouneasly detected buildings. An example of this is shown in Fig. 4.

      In order to solve this issue, the bounding boxes, (width, length and angle) of each building polygon were calculated. Then, all buildings having very long-narrow shapes ($width/length < 0.3$) were removed, which removed these structures without removing buildings that
should be kept in the dataset. This filtering step was verified by visual inspection of the dataset to make sure that only the right objects were removed.

      Finally, since the area of each individual house is hard to estimate correctly from the MLBuildings dataset, the polygon geometries are not needed and therefore we convert the polygon layer into centroid points. The outline area size of each individual building is not used to
calculate the heating energy needs.





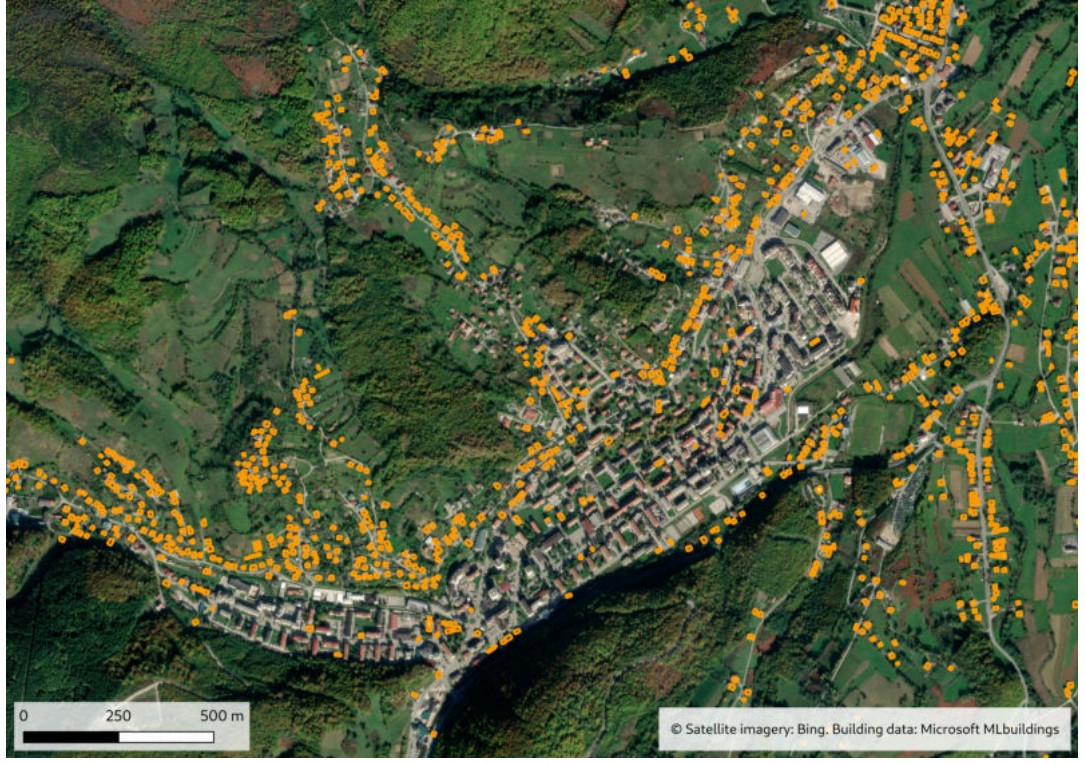

**Figure 3.** Example of the MLBuildings dataset for Novi Travnik (Bosnia and Herzegovina). The image show the same area as in figures 1 and 2 (above), after the buildings have been filtered to only keep 1-3 family housing units. See text for more details.

## 3.2 Energy estimation

In each building point, energy consumption for the different fuels are assigned based on available statistical data. The data provide the energy usage per fuel per 1-3 family housing units. The level of detail, in terms of geographical regions and fuels, varies between the different datasources. Each building point in the dataset is processed to calculate the energy usage.


- Albania: Region-wise energy usage data (3 climate zones), from (Novikova et al., 2015b).

- Bosnia Herzegovina: Entity-based energy usage statistics used, from (Arnautović-Aksić et al., 2016).

- Kosovo: the same data as for Serbia has been used.

- Montenegro: national totals from Eurostat . We assume that all biomass fuels are used for individual housing. The resulting fuel-usage per housing unit is then adjusted for different regions based on degree-days from (Novikova et al., 2018).


- North Macedonia: statistics for each administrative region (Simovski, 2019).

- Serbia: The same data for the whole country have been used, based on (Jovanović Popović and Ignjatović, 2013).

The full table of energy data used is shown in appendix A1.

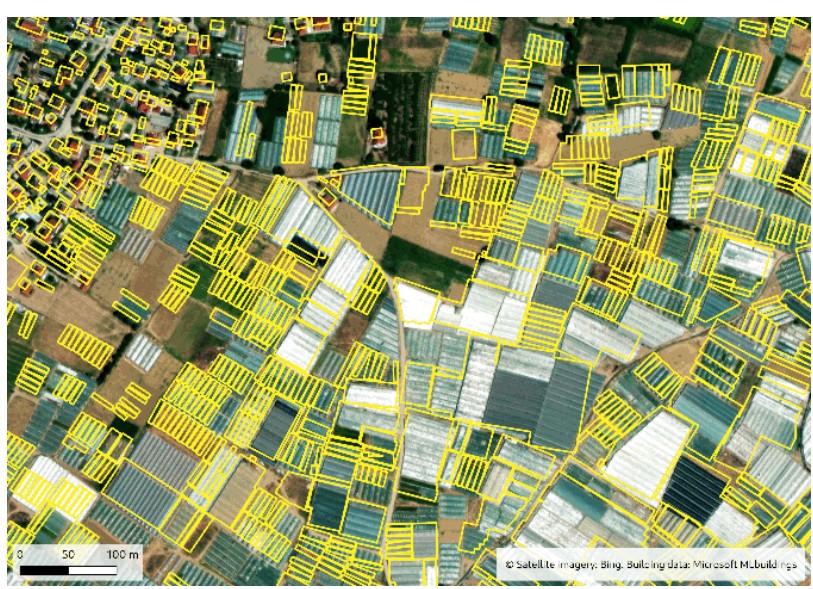

**Figure 4.** False positive building detection on agricultural lands surrounding Strumica, before filtering of such features.



## 3.3 Emissions to air

Once the energy use for each fuel has been assigned to all building points, the resulting emissions for the relevant pollutants may be calculated. The methodology follow the EEA guidebook(European Environment Agency, 2023), for emission factors as well as the composition of heating appliances. For each building point, emissions are calculated for each fuel, based on the composition of heating appliances and the emission factors for each appliance. The following pollutants have been included in the dataset: $PM10$, $PM2.5$, $NO_x$ and $SO_x$.

The MLbuildings dataset contains data based on satellite imagery captured during multiple years, from 2014 and onwards. Therefore the buildings dataset in itself do not correspond to a well-defined year. Similarly, the energy usage data represents multiple years, 2015-2020, while the appliance composition data in the EEA guidebook is for the year 2010. The yearly change in heating emissions from 1-3 family housing units is not expected to vary considerably between the years in the period 2010-2020. We therefore set the year to be 2019, since it is the most recent year in the period that was not affected by the covid-pandemic.

Finally, the point-wise emissions were rasterized into grids for each substance, with 500x500 m horisontal resolution, in the ETRS89-extended / LAEA Europe projection (EPSG:3035). While the calculations of energy and subsequently emissions are done for centroid points for all buildings, the data is rasterized into cells since it might otherwise seem to be more detailed and accurate than it actually is. Further, when using the resulting dataset for atmospheric dispersion modeling or similar, it is computationally more efficient to treat the data as rasters.

## 4 Results and discussion

### 4.1 Buildings per country

Table 1 list the number of buildings together with estimates from other sources. Also listed is the percentage of *urban buildings*, calculated from the number of buildings on urban landuse categories (landuse code $\leq 11240$ in section 2.2 above) The other sources (references are listed in the table) are different statistical reports and not from national databases of building permits or similar. It is also worth noting that in the different references, the categorization of buildings may differ. As far as possible we have compared the number of buildings in this work to 1-3 family housing units in the references.

For several countries the agreement is fairly good, especially for Albania, Republic of North Macedonia and Serbia. For Kosovo and especially for Bosnia and Herzegovina, the methodology used in this work leads to considerably more buildings, 9 and 30 % more respectively. The resulting number of buildings for Bosnia and Herzegovina may be overestimated, though it is uncertain since there is a lack of indendent housing data. It is also uncertain what could cause such overestimation.

Inspection of a random selection of locations suggests that the high number of buildings for Bosnia and Herzegovina is due to the dataset having too many buildings in rural areas where there are smaller farms. This is also supported by the percentage of urban buildings in the dataset, which is considerable lower for Bosnia and Herzegovina (and Montenegro) than for the other countries. This may have a larger effect in Bosnia and Herzegovina than in the other countries. However, the percentage of urban buildings may very well be different in the different WB countries. There are also uncertainties in the references for the number of residential buildings. In order to determine the accuracy of the number of buildings, more independent datasets are required.





| Region | buildings in this work | % urban** | buildings comparison | ref |
|---|---|---|---|---|
| Albania | 593970 | 64.6 | 576096 | (Novikova et al., 2015b) |
| Bosnia-Herzegovina | 1090970 | 49.8 | 841543 | (Arnautović-Aksić et al., 2016) |
| Kosovo* | 420976 | 74.1 | 386046 | (agency of statistics, 2022) |
| Montenegro | 179297 | 55.6 | 158176 | (Novikova et al., 2015a) |
| Republic of North Macedonia | 422274 | 74.4 | 426344 | (Gjorgjievska, 2021; mkl, 2021) |
| Serbia | 2156724 | 74.6 | 2186246 | (Jovanović Popović and Ignjatović, 2013) |

**Table 1.** Total number of 1-3 family houses for the WB countries, obtained from the methodology in this work, compared to other statistical sources. * *All references to Kosovo in this document shall be understood to be in the context of the United Nations Security Council resolution 1244 (1999).* **: percentage of buildings in this work that are on urban landuse categories. See text for more details.

## 4.2 Energy

In figure 5 the total energy is shown. All building points have been rasterized to a 500x500 m grid. It is worth noting some of the the different
patterns that can be seen in this image, in particular one in Serbia: The northern part lies on a plateau and have large agricultural areas with settlements in fairly concentrated villages, leading to a "dotted" pattern in the image. The south-eastern part of Serbia, on the other hand, is more mountainous and have smaller agricultural areas scattered between the mountains and the settlements are also more scattered into smaller pieces. This leads to a more continuous pattern in the image.

When comparing the countries in the Western Balkans in this image, it is worth noting that the calculated result in figure 5 depend on both the number of buildings and the statistical data for heating energy needs.

An uncertainty not included in this study is the fraction of residential buildings that are not in use. This has not been included due to a lack of available data.

## 4.3 Emission data

Maps of emission data for PM2.5, NOx and SOx are shown in figures 6, 7 and 8. Note that the scales in the figures are different. PM10 is not shown, since it is very similar to PM2.5, which is expected for heating emissions due to the fact that most of the particles from combustion in PM10 are fine particles (PM2.5). Both PM2.5 and NOx show similar patterns, while for SOx the emissions in Montenegro, Albania and Republic of Northern Macedonia are much lower compared to the other countries. This is an effect of the heating energy data for different fuels, which is very low or zero for coal in these three countries, leading to very low SOx emissions.

## 4.4 Comparison to other emission datasets

The resulting emission data for NOx and PM2.5 were compared to CAMS-reg 4.2, *sector C* (Kuenen et al., 2021, 2022) as well as EDGAR, *sector buildings* (Muntean et al., 2018)[3]. The CAMS and EDGAR datasets are regional and therefore have a coarser resolution (0.05x0.1 and 0.1x0.1 degrees[4], respectively) than the dataset in this work which is gridded to 500x500 m.

---

[3]Accessed through https://edgar.jrc.ec.europa.eu/country_profile/ , on 2024-09-10.
[4]$0.1x0.1$ degrees corresponds to about 8x11 km for the WB region.





The CAMS emissions were reprojected to the ETRS89-extended / LAEA Europe projection (EPSG:3035) and resampled to 1x1 km grid
size. Then the total emissions for the different regions were calculated using *zonal statistics* in QGIS. The resulting national statistics is
shown in table 2.

| Region | Emission NOx (t) | | | Emission PM2.5 (t) | | | Emission SOx (t) | | |
|---|---|---|---|---|---|---|---|---|---|
| | EDGAR | CAMS | this work | EDGAR | CAMS | this work | EDGAR | CAMS | this work |
| Albania | 1041 | 382 | 796 | 2348 | 3482 | 4103 | 1283 | 499 | 67 |
| Bosnia-Herzegovina | 2182 | 1859 | 5275 | 6755 | 16800 | 46376 | 3852 | 5900 | 20488 |
| Kosovo∗ | - | 479 | 1798 | - | 5461 | 8882 | - | 1135 | 9375 |
| Montenegro | - | 228 | 401 | - | 2008 | 5316 | - | 644 | 88 |
| Republic of North Macedonia | 755 | 644 | 703 | 3269 | 6081 | 7619 | 1047 | 988 | 167 |
| Serbia | - | 2371 | 9212 | - | 23414 | 45505 | - | 9699 | 48028 |
| Serbia and Montenegro | 11197 | 3078 | 11312 | 40055 | 30883 | 59703 | 14771 | 11478 | 57492 |

**Table 2.** Comparison of emission totals for different regions in the Western Balkans, CAMS-regional 4.2 (Kuenen et al., 2021, 2022) EDGAR
(Muntean et al., 2018) Note: for EDGAR the region *Serbia and Montenegro* is still used, which also contains Kosovo∗. The equivalent have
been calculated also for CAMS and this work, to facilitate comparison. ∗ *All references to Kosovo in this document shall be understood to
be in the context of the United Nations Security Council resolution 1244 (1999)*

For NOx, the overall agreement between the two regional dataset and between this work and the regional datasets are similar, though the
results in this work are closer to the EDGAR dataset than to CAMS-reg. The largest difference is found for Bosnia and Herzegovina, where
this work has more than twice the emissions compared to CAMS-reg and EDGAR.


While for NOx the EDGAR dataset has higher emissions than CAMS-reg, for PM2.5 the case is the opposite except for *Serbia and Mon-
tenegro* where EDGAR is about a third higher. For PM2.5 the emissions in this work are considerably higher than both regional datasets.
Since this was not the case for NOx, the discrepancy is not only a result of the number of buildings, but must also be due to different fuel
usage and/or emission factors.


Turning to SOx, finally, the data in this work is rather non-consistent when comparing to the regional datasets. While for some countries
it is much lower, it is considerably higher for other countries. This is a direct effect of the fuel data, as mentioned above. For Albania,
Montenegro and Republic of North Macedonia, there is no coal usage reported (see section 2.3), which lead to very low SOx emissions,
while for Bosnia and Herzegovina as well as Serbia, the coal usage lead to high emissions of SOx, in this case much higher than the regional
datasets.





**Figure 5.** Estimated total energy use for heating in indivudual houses, rasterized to 500x500m. All references to Kosovo in this document shall be understood to be in the context of the United Nations Security Council resolution 1244 (1999)





**Figure 6.** Calculated emissions of for the Western Balkans for PM2.5, rasterized to 500x500m. All references to Kosovo in this document shall be understood to be in the context of the United Nations Security Council resolution 1244 (1999)



**Figure 7.** Calculated emissions of for the Western Balkans for NOx, rasterized to 500x500m.



**Figure 8.** Calculated emissions of for the Western Balkans for SOx, rasterized to 500x500m. All references to Kosovo in this document shall be understood to be in the context of the United Nations Security Council resolution 1244 (1999)





## 5 Conclusions

A complete picture of the air quality situation in a country, region or city requires not only monitoring data, but also an accurate description of the emission sources. Atmospheric dispersion modelling may provide maps of the concentrations for different pollutants and also validate the emission-data. In this work we have developed a dataset for emissions from heating of 1-3 family housing for the whole of the Western Balkans, at high spatial resolution.

While there are multiple sources of uncertainties in the dataset, the basic methodology is

1. obtaining building data, in this case from the MLBuildings dataset

2. filtering of the buildings data, to obtain only the relevant housing units.

3. calculating the heating energy based on available data for energy usage

4. calculating resulting emissions using the EEA emission database

This methodology may be applied wherever the above datasources are available.

Possible improvements in the methodology include better filtering of buildings dataset and, importantly, reliable energy usage data for each region. While the methodology is tailor-made for heating from 1-3 family housing units, other residential housing (apartment buildings) is not included. There are also errors in the MLBuildings dataset, which may be reduced in the future as higher-quality satellite imagery as well as even better building detection AI methods emerge.

Comparing the national totals from the dataset developed in this work with regional emission datasets suggests that the results have acceptable quality and may be used as a basis for emissions analysis, development of air quality action plans as well as input for atmospheric dispersion modeling, which in turn may validate the emission data or lead to improvements in the methodology. It is the hope of the authours that the dataset may help in improving the air pollution situation in the Western Balkans.

*Data availability.* The emission GIS-files (rasters) created in this work are publicly available at https://doi.org/10.5281/zenodo.13906810 (Asker, 2024).





## Appendix A: Energy use per country and region

| Country | region/entity | wood | wood-residue | pellets | coal | lpg | natural gas | oil |
|---------|---------------|------|--------------|---------|------|-----|-------------|-----|
| Albania | climate zone A | 1.2853 | 0.0 | 0.0 | 0.0 | 0.0 | 2.6509 | 0.0 |
| Albania | climate zone B | 1.8538 | 0.0 | 0.0 | 0.0 | 0.0 | 3.0575 | 0.0 |
| Albania | climate zone C | 8.1657 | 0.0 | 0.0 | 0.0 | 0.0 | 7.7732 | 0.0 |
| Bosnia and Herzegovina | Federation BiH | 8.428 | 2.744 | 0.0 | 5.292 | 0.196 | 2.94 | 0.0 |
| Bosnia and Herzegovina | Republika Srpska | 13.054 | 1.926 | 0.0 | 5.992 | 0.428 | 0.0 | 0.0 |
| Bosnia and Herzegovina | Brčko district | 12.524 | 1.212 | 0.0 | 6.262 | 0.202 | 0.0 | 0.0 |
| Kosovo∗ | - | 3.93 | 0.0 | 0.0 | 6.71 | 0.0 | 4.23 | 1.71 |
| Montenegro | climate zone I | 7.26 | 0.52 | 0.26 | 0.0 | 0.0 | 0.0 | 0.0 |
| Montenegro | climate zone II | 11.31 | 0.81 | 0.41 | 0.0 | 0.0 | 0.0 | 0.0 |
| Montenegro | climate zone III | 15.16 | 1.08 | 0.55 | 0.0 | 0.0 | 0.0 | 0.0 |
| Republic of North Macedonia | Northeastern | 7.10 | 0.00621 | 0.671668 | 0.0 | 0.006869 | 0.07089 | 0.0124768 |
| Republic of North Macedonia | Skopje | 3.02 | 0.01081 | 0.410650 | 0.05500 | 0.154395 | 0.0 | 0.111276 |
| Republic of North Macedonia | Southwestern | 10.32 | 0.01127 | 0.547852 | 0.0 | 0.078288 | 0.0 | 0.022153 |
| Republic of North Macedonia | Eastern | 11.04 | 0.00575 | 0.413996 | 0.0 | 0.118660 | 0.0 | 0.101164 |
| Republic of North Macedonia | Southeastern | 8.18 | 0.0 | 0.329859 | 0.0 | 0.045015 | 0.42060 | 0.048908 |
| Republic of North Macedonia | Pelagonia | 6.33 | 0.0 | 0.845203 | 0.0 | 0.092912 | 0.0 | 0.0 |
| Republic of North Macedonia | Vardar | 6.36 | 0.0 | 1.047899 | 0.0 | 0.060023 | 0.0 | 0.0 |
| Republic of North Macedonia | Polog | 8.59 | 0.00253 | 1.324693 | 0.050417 | 0.096611 | 0.0 | 0.282571 |
| Serbia | - | 3.93 | 0.0 | 0.0 | 6.71 | 0.0 | 4.23 | 1.71 |

**Table A1.** Energy usage ($MWh/y$) per residential building for each country and region. For Albania the regions follow (Novikova et al., 2015b) while for Montenegro (Novikova et al., 2015a). ∗All references to Kosovo in this document shall be understood to be in the context of the United Nations Security Council resolution 1244 (1999)

*Author contributions.* CA: conceptualization, methodology, data processing, analysis, visualization, writing. EvD: methodology, verification of emission calculations, writing. OT: project managemeng, writing

*Competing interests.* The contact author has declared that none of the authors has any competing interests.

*Acknowledgements.* The authors would like to thank the Swedish Environmental Protection Agency (SEPA) and environmental agencies in the Western Balkan countries for collaboration.





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
