# Peer review of "Residential heating emissions for the Western Balkans"

_Earth System Science Data, 2024_

## Author Response (AR1)

Reply to teviewer 1:

We thank the reviewer for the positive and constructive comments.

Regarding reviewer comment 1 and 2:

The selected fuels are based on the available fuel usage data for the Western Balkan countries and regions.

For some of these countries, such as Albania, the fuel usage data do not contain wood residue at all. However, it is likely there is at least some wood residue used also in Albania, but we lack information about that.

Similarly, for Bosnia and Herzegovina, e.g. ,there is fuel data for wood residue but no clear definition of what the wood residue contains, such as straw or corn-hobs. For clarity, this is explained in the updated version of the manuscript.

Further comment:

We also agree that combustion in agriculture can be an important source of emissions.

Part of our methodology aims to discriminate the 1-3 family houses in all areas, as parts of the residential heating sector. For smaller farms that lie on the outskirts of residential areas, correct detection of housing units with our methodology is difficult, which is mentioned in the manuscript. However, there may also be uncertainties in how much of the statistical fuel usage data is for heating of housing and for heating of buildings related to farming. This makes the distinction between emission sectors difficult in such areas. To improve this issue, better methods for detecting the housing units for smaller farms is needed, as well as more detailed fuel usage data. While this issue constitute an uncertainty in our methodology, we believe that the method presented in the manuscript is a first step towards more detailed emission data for the region and a first step that can serve as a basis for further elaboration and improvements.

Reply to reviewer 2:

We thank the reviewer for the comments.

From the scope of the ESSD journal:

"Earth System Science Data (ESSD) is an international, interdisciplinary journal for the publication of articles on original research data (sets), furthering the reuse of high-quality data of benefit to Earth system sciences."

Based on this, we believe that the manuscript and corresponding dataset falls within the scope of the journal.

- While the innovation lies in the the use of AI-based building detection data combined with other open dataset to obtain a dataset of individual housing units, we agree that the innovation of the methodology can be more clearly described and emphasized.
- Similarly, while the importance of the manuscript and dataset is demonstrated, it may be more clearly described and emphasized.

Both these issues has been improved in the new version of the manuscript.

---

## Author Response (AR2)

Reply to the editor:

We thank the editor for the constructive comments.

The suggestions from the editors are:

Therefore, please add a more details on the methods used to produce your results.
 Additional minor points:
 1. Capitalize the word "Figure" in the text (instead of 'figure") whenever this is pointing to a Figure in the manuscript.
 2. Please add text to explain why the boundaries of some countries in Figs 5-6-7-8 appear to extend to sea water

Numbered points 1 and 2 has been resolved in the new version.

Regarding details on the methodology:  we have added some details about the methodology. We have also added sub-sections to make the methodology section clearer and easier to read and re-arranged some text paragraphs.

While every detail of the methodology for all substances/fuels etc would make the manuscript very long and tedious to read, we have added some examples to make the methodology easier to follow.